# From Regional Dialects to the Standard: Measuring Linguistic Distance in Galician Varieties

**Xulio Sousa** 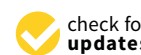

Instituto da Lingua Galega, Universidade de Santiago de Compostela, 15782 Santiago de Compostela, Galicia, Spain; xulio.sousa@usc.es

**Abstract:** The analysis of the linguistic distance between dialect varieties and the standard variety typically focuses on establishing the influence exerted by the standard norm on the dialects. In languages whose standard form was established and implemented well after the beginning of the last century, however, such as Galician, it is still possible to perform studies from other perspectives. Unlike other languages, standard Galician was not based on a single dialect but aspired instead to be supradialectal in nature. In the present study, the tools and methods of quantitative dialectology are brought to bear in the task of establishing the extent to which dialect varieties of Galician resemble or differ from the standard variety. Moreover, the results of the analysis underline the importance of the different dialects in the evaluation of the supradialectal aim, as established by the makers of the standard variety.

**Keywords:** galician language; dialects; standard; typology; linguistic distance; dialectometry; language planning; language standardization

## 1. Introduction

The measurement of linguistics differences is an old topic of linguistic research that has been given new life in recent decades by the advance of data-intensive computing systems in linguistic research. The two main topics of interest in this type of study are the measurement of the linguistic distance between language systems (languages of different families, languages of the same family, social varieties, regional varieties, etc.) and the analysis of linguistic differences within language systems, especially in computational linguistics (word-sense induction, word-sense disambiguation, etc.). In the last three decades, the first type of research has been particularly fruitful in areas such as language typology, forensic linguistics, areal linguistics, historical linguistics, comparative linguistics, language variation, and even literary studies (Cysouw 2005; Borin and Saxena 2013; Jenset and McGillivray 2017). The extension of the use of computational methods in research and the increasing availability of large linguistic data set in digital form has facilitated the work of researchers and has contributed to increasingly more reliable and robust results. Indeed, in the last few years the most popular application area of this analysis perspective is dialectology, the study of regional variation in language. Within this discipline, a set of methods of statistical analysis and visualisation of cartographic information has been set up which is associated with the names of *quantitative dialectology*, *aggregate dialectology* and, most commonly, *dialectometry* (Goebl 2006, 2008; Szmrecsanyi 2014; Goebl 2018). This new set of methods and techniques of analysis allows for a more complete and grounded knowledge of the spatial organisation of linguistic variation. As Borin and Saxena (2013, p. 59) points out, current dialectometric studies may be considered as part of research in genetic linguistics, insofar as the methods used to measure the distance between dialects are the same or very similar to those used to determine the distance between languages and families of languages.

The procedure used in dialectometric analysis is based on the handling of large amounts of data and the use of quantitative methods to exploit information (Nerbonne 2006; Dubert and Sousa 2016). The combination of these foundations allows researchers to overcome the limitations of methods used in traditional dialectology. The linguistic space can be studied taking into account all the information provided by the geolinguistic studies and consequently the results of the analysis will be more robust and reliable, as well as being easier to contrast with information extracted from other sources (genetics, history, demography, geography, etc.). Each spatial entity (individual, locality, region) is characterised by the set of variables under research (phonetic, morphological, lexical, etc.). The data obtained from the analysis make it possible to identify similarity, distance and correlation relationships that using traditional dialectology methods are impossible (Goebl 2008; Szmrecsanyi 2014).

The dialectometric method is commonly applied to study geolinguistic varieties in an extensive or a limited linguistic domain but can also be used to analyse non-geographic varieties and even to compare, as in the current study, standard varieties with real varieties spoken at any given time and at a precise location (geographical lects).

In recent years, several studies on the Romance linguistic space have been concerned about exploring the distance between the standard variety of a language and the regional varieties of that language using quantitative methods. Hans Goebl studied the influence of standard French and Italian varieties on the local varieties of border areas of France and Italy (Goebl 2000). Valls used the standard variety of Catalan as a reference to analyse phonological and morphological changes in north-western Catalan (Valls 2013). In the Italian domain, the lexical differences between the Tuscan dialects and the Italian standard have been analysed, and in this case taking into account linguistic, geographical and socioeconomic factors (Wieling et al. 2014).

The current study is part of this line of dialectometric research centred around the analysis of linguistic distances between the so-called standard variety of a language and the set of regional varieties of a linguistic domain, in this case the Galician linguistic domain. Despite the proximity of this study to some of the previous research, this analysis differs from other comparative studies between linguistic varieties in two ways.

Firstly, it focuses on calculating distances between varieties to discover in what ways the standard variety comes close to or distances itself from spoken forms of the language. Most studies that analyse such questions seek to identify influences of the standard on the dialects (Bellmann 1998; Gerritsen 1999; Wolfram and Schilling 2015; Valdman et al. 2015; Cerruti et al. 2017). However, mainly because Galician standardisation is so recent, the current paper adopts an approach that helps identify to what degree the different dialectal varieties have contributed in shaping the standard. The impact that endoglossic standard has had on the structure of regional dialects, use and status, falls outside the purposes of this paper.

Secondly, this approach also differs from previous works about Galician standard in terms of the method and data used. Within Galician studies, allusions have sometimes been made to the closeness of the standard language to certain Galician dialect varieties. Such studies generally focus their attention on a limited number of features, rarely including more than half a dozen morphological and phonetic features (Regueira and Luís 1999; Fernández Rei 2007; Beswick 2007). Consequently, the suggested results and conclusions are inconsistent. The current research is based on the premise that a fuller analysis of linguistic variation, and, specifically, of similarities and differences between varieties basically requires the analysis of wide range of linguistic variables. Only in this way can conclusions be reliable and robust.

This article is structured as follows: firstly, it presents the background by providing an overview of the situation of Galician and the conformation of its standard Secondly, information about the data studied and the method used in the analysis is provided. In a next step, the results obtained are connected to previous studies that addressed the same issue. The conclusion presents some reflections on the results of the research together with some considerations on the usefulness of aggregate data analyses as a way to evaluate the distance between language varieties.

## 2. Galician Language: From Dialects to Standard

The Galician language is a Romance variety spoken in a geographical area that includes present day administrative Galicia (northwest Spain) and border areas of Asturias, León and Zamora (Figure 1). The varieties of Latin spoken in the northwest of the Iberian Peninsula are the fundamental sources that have given rise to the languages that are currently identified as European Portuguese and Galician (Dubert and Galves 2016). Galicia is an autonomous community that belongs to the kingdom of Spain and therefore it is easy to understand that Galician and Spanish have lived in the same administrative territory for centuries. Today, the majority of the Galician population states that they understand and speak Galician and Spanish without difficulty, although the percentages of daily use of the two languages are not identical and have changed markedly in the last decades to the detriment of Galician (Monteagudo Romero 2017). The recognition of Galician as the official language and the beginning of the socialisation of its standard variety was not achieved until the last third of the last century. As a result, Galician has long been a language spoken by the majority of the population living in a situation of inequality of use and social recognition with Spanish, the language of administration, education and the media. In the words of Darquennes and Vandenbussche (2015, p. 2), today "Galician is an autochthonous language minority in a situation of asymmetrical language contact."

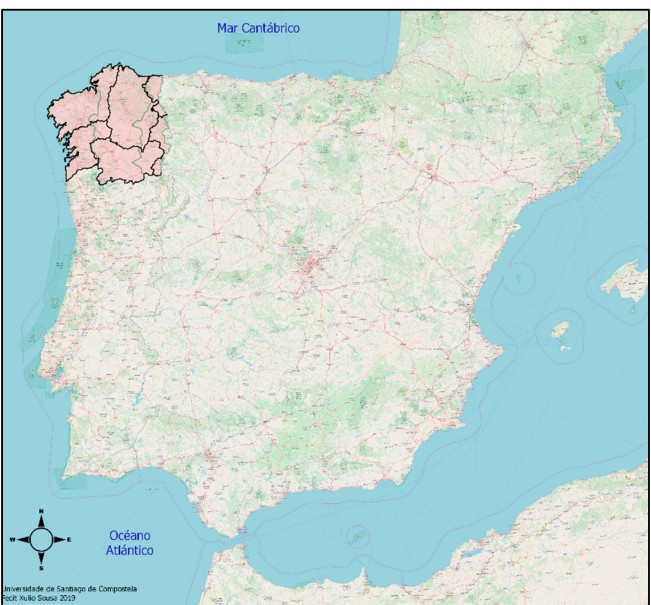

**Figure 1.** Galician linguistic domain in the Iberian Peninsula.

### 2.1. Galician Dialects

From the point of view of linguistic studies, the recognition of Galician in the panorama of Romance studies was very much determined by the historical and political weight of the nearby official languages (Spanish and Portuguese), whose standard varieties had been fixed and socialised since the beginning of the modern age. In addition, most scholars had very little knowledge of the real situation of the language in the Galician language domain. In the classical textbooks of Romance linguistics, Galician appeared either as a variety of Portuguese (Diez 1882; Tagliavini 1949; Meyer-Lübke 1926; Lausberg 1965) or as a dialect of Spanish (Vidos 1967). A more precise characterisation of the linguistic system and a more up-to-date view of the language situation can be found in the works published from the end of the 20th century, in which Galician is already recognised as an autonomous linguistic variety with close historical relations with European Portuguese (Holtus et al. 1994; Harris and Vincent 1988; Posner and Green 1993; Pöckl et al. 2004; Monjour 2014; Dubert and Galves 2016).

The dialectal variation of contemporary Galician is known in detail thanks to two geolinguistic projects developed during the 20th century. The *Atlas Lingüístico de la Península Ibérica* (ALPI),

a documentation project of the Romance varieties of the Iberian Peninsula in the first half of the century, offers information on rural Galician spoken in sixty localities of the Galician linguistic domain (García Mouton et al. 2016). The *Atlas Lingüístico Galego* (ALGa) project documented the Galician regional varieties in the seventies of the same century in 167 localities. Both are linguistic atlases, designed from presociolinguistic dialectology, and they present a language variety that responds to the characteristics of NORM speakers, as proposed by Chambers and Trudgill (1998).

The materials collected for the ALGa project have already been published for the most part and served to carry out a detailed description of the regional variation of Galician, both by applying the methods of traditional dialectology (Santamarina 1982; Fernández Rei 1990b), and from perspectives using more updated quantitative methods (Sousa 2006; Dubert García 2011). These studies are based on the analysis of the phonetic and morphosyntactic data of the ALGa, since the lexicons have specific characteristics, both for the greater number of variants per variable and for their territorial distribution (Sousa 2017). These proposals coincide in identifying three major dialectal areas separated by boundaries running from north to south: western area, central area and eastern area (Figure 2). These studies also agree in pointing out the high linguistic uniformity existing in the linguistic domain and the absence of geographical varieties that are perceived and recognised by the speakers themselves as such (Fernández Rei 1990b). In addition, it should be borne in mind that this fragmentation within the Galician domain follows the organisation pattern of the primary or constitutive Romance dialects in the north of the peninsula, pointed out by several authors. Penny (2003, p. 76) recognises the existence of a "northern Peninsular dialect continuum" which extends along the north third of the peninsula, and "is part of the Romance dialect continuum which extends from north western Spain into France and thence into Belgium, Switzerland and Italy" (Gargallo Gil 2011; Ossenkop 2018a, 2018b).

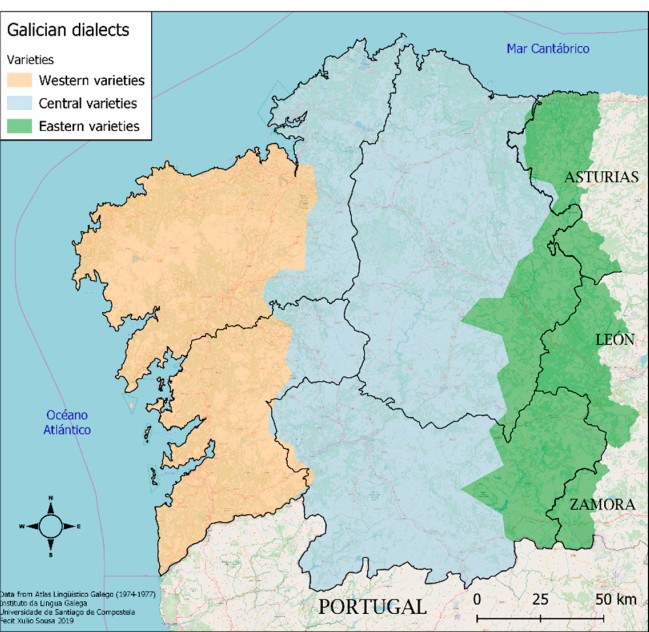

**Figure 2.** Galician dialectal regions.

## 2.2. Galician Standard Variety

The process of fixing the standard variety of Galician was carried out in the second half of the twentieth century, with quite a delay as regards Spanish and Portuguese. Although Galician was the language used by the majority of the population, attempts to provide it with a standard written variety were unsuccessful due to the lack of essential support from the government. For centuries Spanish was the only language with official recognition and therefore also the only one used as a model in education, official communications, written culture and religious offices. In this sense, Galician was in a similar situation to that of other minority languages of Spain.

In 1981 Galician is recognised as the co-official language of Galicia. As a result, it was essential to have a standard variety of language that could serve as a template for the new areas of use (teaching, administration, media, etc.) and for it to have a written form suitable to the new functions.

Unlike what happens in other languages with older and more lasting coding processes, in the case of Galician, there is sufficient available information to reconstruct the principles on which the standard was developed. The task was carried out by the Real Academia Galega, an institution created in the early 20th century, following the example of the Real Academia Española, and by the Instituto da Lingua Galega, a university research centre dedicated to the study of Galician language since the early 1970s (Monteagudo Romero 2017). The proposed language model differed little in essence from what had been used in Galician literary and non-literary writing since the mid-20th century (Monteagudo Romero 2003). This process of selection was not without debate between defenders of two normative currents: one that maintained the tradition consolidated from end of the 19th century of a model of language based on the modern spoken varieties and another one that tried to establish a variety identical or very near the standard of European Portuguese. The first of these proposals was the one that was consolidated and finally ratified by the academic and institutional bodies (Fernández Salgado and Romero 1995; Ramallo and Rei-Doval 2015).

This standard variety model was based on the idea of language as a symbol of identity, insofar as Galician was intended to be understood by speakers as a distinctive language as regards the closest varieties, Portuguese and Spanish (Kloss and Haarmann 1984). The normative code of this variety was made explicit in the *Normas ortográficas e morfolóxicas do idioma galego* (hence forth *Normas*; (RAG and ILG 1982), a publication that functions as a language codex, in the sense defined by Ammon (2003, pp. 4–5): it contains "the result of codification" and serves as a "guide for correct language behavior or of corrections of language behavior". This work establishes the standard variety model as to spelling and basic grammatical aspects. Scholars in the process of establishing the Galician language standard point to four fundamental criteria that guided the codification and which are contained in the same text of the *Normas* (Monteagudo and Santamarina 1993, pp. 157–58):

1.　internal coherence and paradigmatic economy, with the flexibility to admit alternative forms with a tradition in modern literary Galician;
2.　vitality of the forms in popular Galicia, giving preference, between forms which are equally suitable according to the other criteria, to those that are most widely diffused or used by most people;
3.　use in modern literary Galician, giving preference to classical authors but taking into account the main tends in the evolution of the written language;
4.　linguistic autonomy, enabling Galician to be identified as clearly distinct from Castilian, which implies selecting, from among forms equally permissible according to other criteria, those which are different from Castilian; and in harmony with other Romance and West European languages, particularly with Portuguese.

Out of the four criteria the one which undoubtedly proved most decisive in the selection of the morphosyntactic characteristics of the Galician standard was the second: The text of the *Normas* points out that the standard variety was not to be based on a single dialect, but had to be supradialectal:

> Standard Galician must be a common vehicle of expression valid for all Galician people, as an apt and accessible voice for their written and oral, artistic and everyday manifestations alike. Consequently, common Galician cannot be based on a single dialect, but must give preferential consideration to the geographic and demographic scope of forms when choosing those that are standard. Hence it is to be supradialectal and should aim to achieve acceptance of the solutions adopted by the greatest possible number of Galicians.
>
> (RAG and ILG 2004, p. 11)

The principle of proximity of the standard to the spoken varieties also governed some of the coding proposals that emerged in the late 19th and early 20th centuries (Monteagudo Romero 2003).

This standard language model is described as a *transdialectal* and *compositional* variety. It is a result of the kind of standardisation that Haugen (1972) describes as a compositional thesis of selection. According to Deumert and Vandenbussche (2003, p. 5) this is the model followed for the configuration of most European language standards, since they are composite varieties that were formed over time and gathered characteristics of several dialects.

The variety proposed in the text of the *Normas* is what was implemented as a written language model and is therefore widely used in education, official documentation, major publishers and the media. In 1995 and 2003 the text of the Normas was modified in minor aspects that did not affect the principles that govern the codification (Monteagudo Romero 2003).

## 3. Data and Method

### 3.1. Data Sources

The data used in this study come from two sources. The first is the latest version of the text which presents the graphical and morphological features of the standard variety (RAG and ILG 2004). The second is a project on the language geography of the Galician linguistic domain that started in the 1970s, and whose results began to be published in 1990, the *Atlas Lingüístico Galego*. This work was the source of information provided in the first two published volumes: verb morphology (Fernández Rei 1990a) and nominal morphology (Álvarez Blanco 1995). The language atlas was designed and developed in line with the most traditional procedure in geolinguistics, surveying different places within the area that in Romance philology is considered to constitute the Galician language domain. This area extends beyond the eastern border of the administrative territory of Galicia. In all, information from 167 locations was collected. It is a linguistic atlas that deals with the rural variety spoken by low-mobility adults, NORMs under the name proposed by Chambers and Trudgill (1998).

The features analysed pertain to the area of morphosyntax, understood in a broad sense to include not only rules of inflection and word formation, but also variation in the form of grammatical words (Table 1). In a standard variety that is still being consolidated such as that of Galician, it is far more difficult to select phonetic, prosodic or lexical features, since they are language levels in which it is not always easy to identify the variants to be considered as models.

**Table 1.** Example of variables (features) and variants studied in this paper.

| -óns (plural of nouns ending in -ón) | catro 'four' | ti 'you (sg.)' | sei 'I know' | ningunha 'none f. sg.' |
|:---:|:---:|:---:|:---:|:---:|
| -óns<br>-ós<br>-ois | catro<br>cuatro | ti<br>tu | sei<br>sein<br>sen | ningunha<br>ninguna<br>ningúa<br>niñuna |

The selection procedure consisted in going through the map index of the two volumes of ALGa and picking out maps that met four conditions:

i.　　the studied variable is included in the text of the *Normas*;
ii.　　the map provides information for all localities in the territory;
iii.　　one of the variants shown on the map matches the standard form proposed in the *Normas*;
iv.　　the map shows the existence of variation in the Galician domain, that is, at least two variants for each variable.

In the interpretation of the data it is necessary to bear in mind the latter condition, since it has as a consequence an over-dimension of the differences existing between the dialectal varieties analysed. The principle was adopted because linguistic atlases are some of the fundamental sources of the research and these works offer information on the variables.

After applying the restrictions on the map set of the two volumes of the ALGa explored, a corpus of 324 variables was obtained, which are the ones analysed in this study.

*3.2. Method*

The data for each map were classified in such a way as to identify variables which enabled a comparison according to the above principles. The variants that occur in ALGa were recorded, for each variable (maps from the ALGa), as was the one found in standard Galician; this gave a total of 168 records per variable, 167 corresponding to the places covered by ALGa plus one for the standard variety. This information is a data matrix with the dimensions N (inquiry points) × p (variables or ALGa maps analysed). From this duly classified and ordered set of data, the matrices of similarity and distance between the elements under analysis were calculated: the localities researched in the ALGa and the standard variety, considered in the database as an artificial locality. The similarity index used was that proposed by the Salzburg school of dialectometry: Relative Identity Value ($RIV_{jk}$). This index has proved to be very helpful to "measure the percentage of pairwise matchings between the discrete nominal (or qualitative) types of those linguistic features" (Goebl 2010, p. 65). The results of this calculation are transferred to a squared similarity matrix (N × N) which leads to knowing the similarity (or distance) between the analysed 168 points. In order to enhance the interpretation of the data obtained in the analysis, a choropleth map consisting of coloured polygonal shapes was used[1].

The software used to classify and process all the linguistic data were *Visual DialectoMetry* (VDM), designed by Hans Goebl and created by Edgar Haimerl, and the programming language R, a software environment for statistical computing and graphics (R Core Team 2019). The maps were created using QGIS, an open-source desktop geographic information system (GIS) application (QGIS Development Team 2019), and VDM.

**4. Results and Discussion**

Dialectometric techniques consider linguistic variation quantitatively, allowing to obtain information about the spatial stratification of the dialectal similarities as regards the standard. The histogram in Figure 3 shows how the similarities between the standard variety and the set of localities investigated in the ALGa are distributed from the 324 morphosyntactic variables analysed. On the horizontal axis, the ALGa points are ordered grouped from left to right, from the lowest similarity (33.4%) to the highest (89.02%). The average degree of similarity is 74.75%, the median 76.52%, and the value that appears most often 84.76%. The 167 values are grouped using the natural breaks classification method (also called Jenks optimisation method), very popular in geostatistics. The points are clustered into four classes and each class is assigned a single colour: cold colours for points with lower similarities (blue) and warm colours for the higher ones (red). This classification method seeks to minimise the variance within classes and to maximise the variance between classes. The map of Figure 4 presents each class distribution in the territory. The points linguistically less similar to the standard variety (blues) are located on the eastern edge of the territory. The darker colours mark the greatest distance from the standard and correspond to points of Asturias and León, which share linguistic features with the Asturian-Leonese dialects. These include the locality that has the least similarity to the standard variety, Pombriego, in the province of León (Le-05).

---

[1]　The base map was created using the methods of Delaunay triangulation and Voronoi polygonisation.

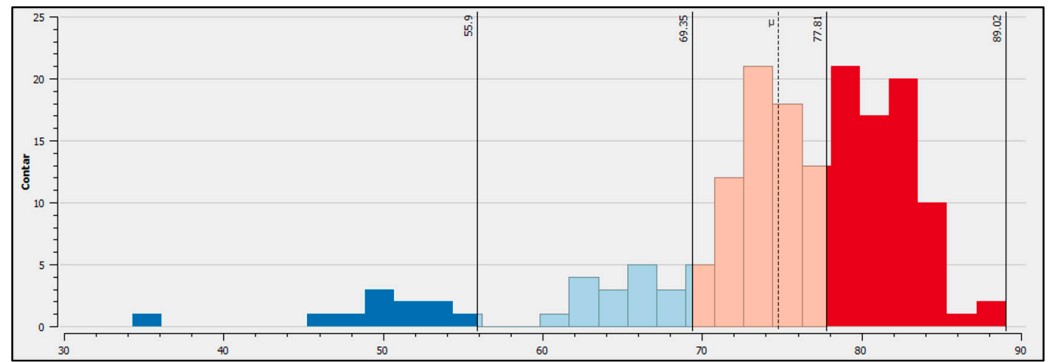

**Figure 3.** Histogram of similarity as regards the standard variety.

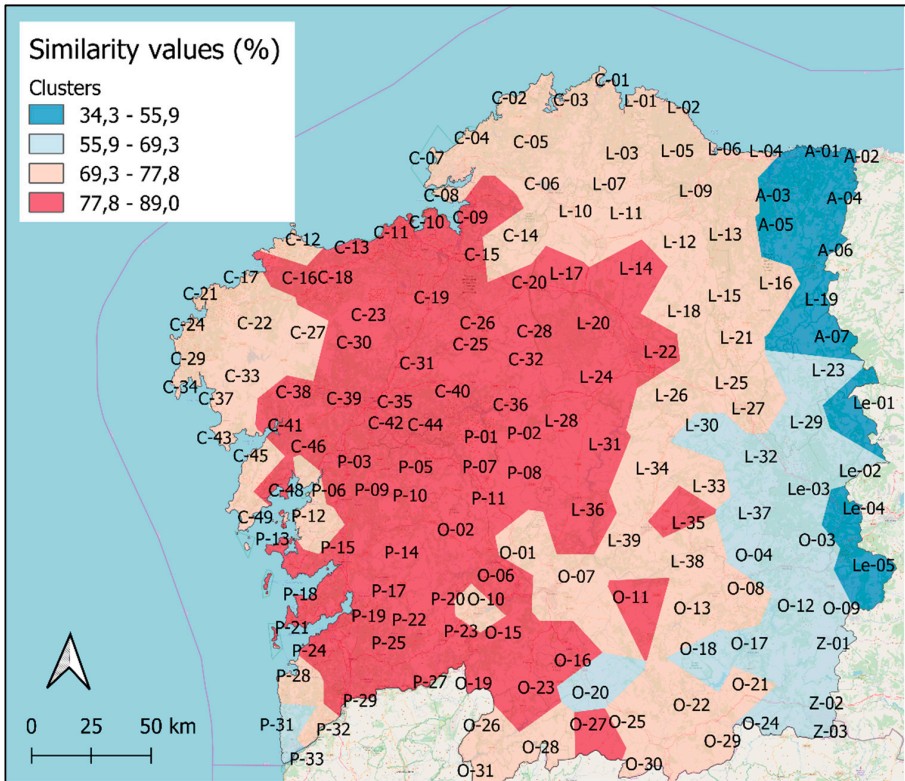

**Figure 4.** Similarity map: spatial distribution of the similarity values refering to standard variety.

In dark red, points showing the highest similarity values (between 77.8% and 89.02%) are highlighted. They occupy an region that lies at the border of the western and central dialectal areas, extending further into the latter, especially in the provinces of A Coruña and Pontevedra. Among these points is the locality of A Lama (P-17), which has the highest similarity index (89.02%). The distribution in the territory of this red area can be interpreted as signalling that solutions adopted in the standard variety sometimes coincide with the western dialects and sometimes with the central dialects.

The second analysis aims to situate standard Galician vis-à-vis the three main dialects traditionally recognised in Galician. The statistical procedure used was that of cluster analysis. The 324 morphosyntactic variables were analysed for 168 points (167 localities of the ALGa and the artificial point corresponding to the standard variety). In this case clustering involves gathering data points according to values of similarity. The agglomerative procedure used was Ward's method, which has already been effectively applied in dialect studies (Goebl 2018). The three clusters resulting from the analysis of the data considered in the current study are shown in the map of Figure 5 and the dendrogram of Figure 6. The areas distribution is very similar to the traditionally acknowledged dialect division, with three basic dialects which display

small differences in their border zones (Fernández Rei 1990b). The artificial point corresponding to the standard variety is identified on the map with the number 500 (standard Galician, south of the border between Galicia and Portugal on the map). This point appears with the same green colour as the points of the western zone, that is, the linguistic features of the standard variety are closer to the western varieties of Galician. In other words, because of the number of variables analysed herein, the standard variety belongs to the area of western dialectal varieties of the Galician domain.

In some previous works based on the consideration of a very limited number of variables, assessments are made on the similarity of the standard variety and the Galician dialects. In a study which runs through the history of the standard of the contemporary Galician, when evaluating the weight of the dialects in the standard variety, Santamarina indicates that in the choices it is possible to note "a certain western accent", which is justified by the greater demographic and literary weight of western Galicia (Santamarina 1995, pp. 78–79). Pountain, when dealing with standardisation processes in the Romance languages, emphasises that Galician is an example of an *eclectic standard* in which, in addition, the purpose of distancing Portuguese and Spanish has played an essential role in the construction of the standard. This scholar points out that the standard of Galician "has tended for the most part to follow the pronunciation of the central vernaculars but the morphology of the southwestern vernaculars" (Pountain 2016, p. 637). The similarities of the standard variety with the dialects spoken in the southwestern area of Galicia are also noted by Beswick. This author reinterprets the words of Santamarina (1995) and indicates that the morphology of the standard is based on the dialects "spoken in the Iria Flavia and western regions"; whereas the phonological standard "was termed Galician *lucense*, spoken in the areas of Lugo, Mondonhedo and Ourense" (Beswick 2007, p. 133). In a more recent text, dialectologist Fernández Rei reviews the weight of dialects in the setting of a standard and states that "the central block dialects are the ones best represented in both phonetics and morphology" (Fernández Rei 2013, p. 75) of this variety.

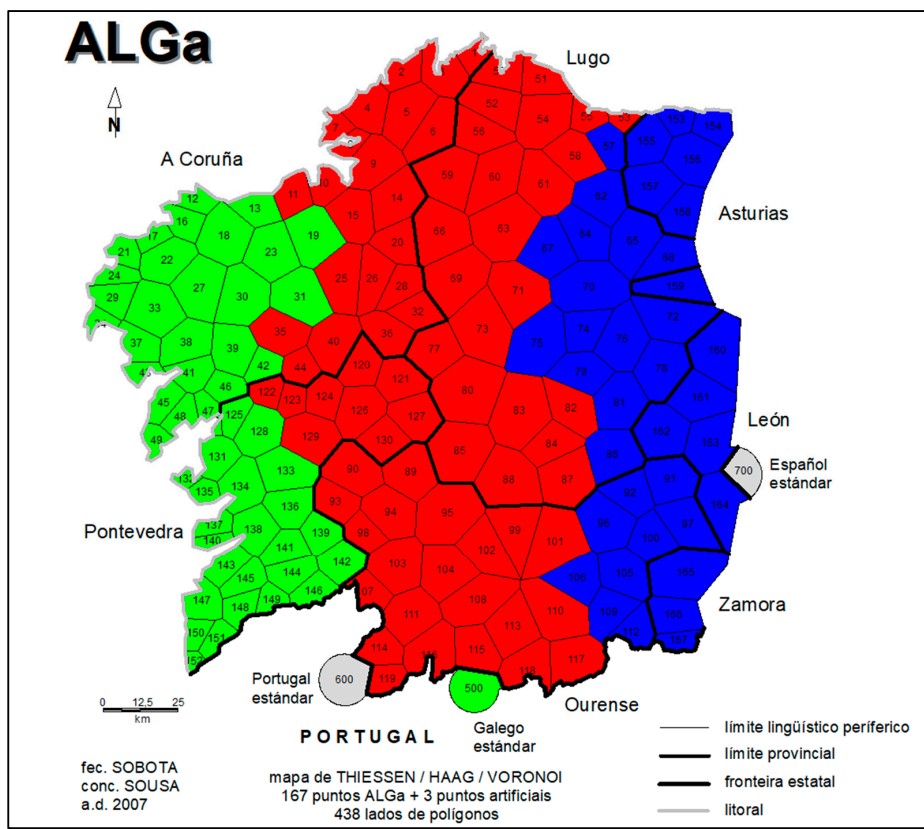

**Figure 5.** Cluster map: dendrographic classification of 168 dialectological points (324 morphosyntactic features).

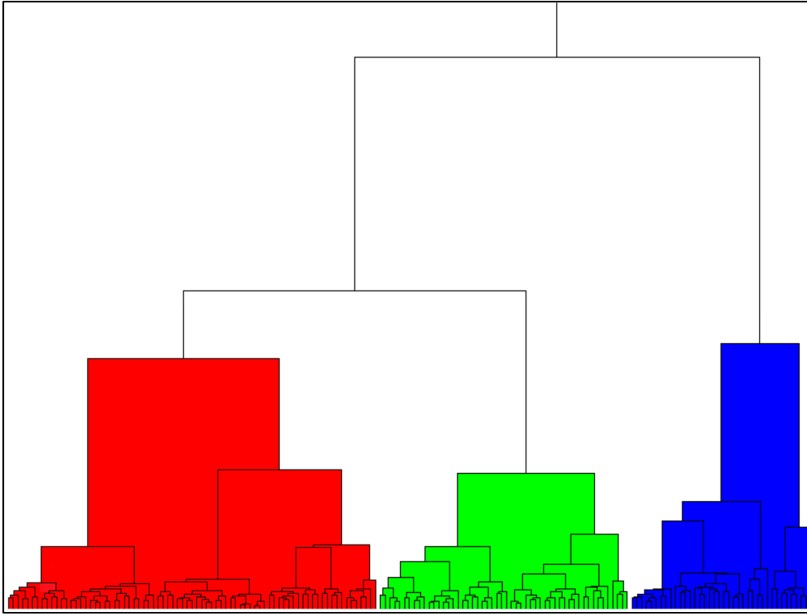

**Figure 6.** Dendrogram: dendrographic classification using Wards's method (3 clusters).

## 5. Conclusions

In recent years studies on the relationships between language varieties and standard variety have focused on the analysis of levelling processes, as these are the ones that most directly affect the most studied western languages (Auer 2018). The long history of most standard varieties of European languages justifies the interest of linguistics in knowing in what direction the influences between varieties take place. On the one hand, in countries such as France, Portugal and Denmark, socialisation of the standard has led to the practical disappearance of regional varieties. In others, such as Germany and Switzerland, dialects are much more resistant to the influence of the standard. In language domains with well-established standards and long tradition, quantitative analyses of distances between dialects and the standard variety help to find out the similarities and differences between these linguistic forms. In addition, they are a useful scientific method for measuring how convergence, divergence, and levelling movements occur.

In those languages in which the standard has a shorter history and a weaker socialisation, the analysis of relationships between varieties can be done by adopting other perspectives. In this article, the study of the linguistic distances between rural geographical varieties of Galician spoken at the end of the 20th century and the standard norm of this language was addressed. The application of multidimensional statistical methods has enabled us to obtain an overview of the similarities between the standard and the variety set, which may be evaluated more rigorously once similar studies on other language varieties take place. The conclusions that can be drawn from this contribution to the contrastive analysis between the standard variety and the geolinguistic varieties of Galician are fundamentally two:

1. The first is that, overall, we find a high degree of similarity between the standard variety and the different geographical varieties. In more than half of the places surveyed, the similarity index is higher than 76%. In this sense, the data may be said to show strong affinity between the standard language and the dialect varieties. Standard Galician is seen, in this analysis, to exemplify the compositional standard model.

2. A second conclusion concerns the distribution of similarity and confirms what scholars had already stated intuitively: the places that display the highest degree of similarity with the standard variety are those located in the central-western part of the Galician language area. It is possible to add that if the standard variety were one of the regional varieties it would form part of the traditional western area.

The results speak for themselves and are useful for filling out and improving the intuitive descriptions that were available until now. They also show that the corpus used in this study seems to be adequate to bear out both conclusions. It would be interesting if in the future these findings could be compared with an analysis of the perceptions of speakers themselves regarding the differences between their speech variety and the standard.

**Funding:** This research was funded by the State Programme for the Development of Scientific and Technical Research Excellence, State Subprogramme of Knowledge Generation of the Spanish Ministry of Science, Innovation and Universities, grant number PGC2018-095077-B-C44 (MCIU/AEI/FEDER, UE).

**Acknowledgments:** I wish to express my gratitude to Adela Martínez Calvo for her valuable advice in the statistical analysis.

**Conflicts of Interest:** The author declares no conflicts of interest.

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
