# Peer review of "From Regional Dialects to the Standard: Measuring Linguistic Distance in Galician Varieties"

_languages, doi:10.3390/languages5010004_

Round 1

Reviewer 1 Report

42-43 The combination of these foundations allows researchers to overcome the limitations of the intuitive and partial methods used in traditional dialectology

→ The combination of these foundations allows researchers to “offer a parallel way to traditional methods used in” dialectology

(otherwise you are stating that trad. dialectology does not use scientific methods)

49-50 that using traditional dialectology methods are impossible

consider reformulation

130 responds to the characteristics of NORM speakers.

consider adding more information (e.g. “as proposed by Chambers and Trudgill (1998)”)

147 Gargarllo Gil

→ “Gargallo”

230 spoken by low-mobility adults, NORMs under the name proposed by Chambers and Trudgill

consider reformulation

238-244 present selection criteria and may be anticipated before the choices at lines 232-237

273 Dialectometric analysis of the data makes it easy to obtain all the information about the spatial

consider reformulation

274-275

clarify:

The histogram in Figure 3 shows the number of points of the ALGa within the uniform classes of similarity with the standard”? Steps are about 2%? Are they automatically defined by an algorithm?

Explain how the number of clusters is decided and how it may be forced to three in the following lines...

303 according values of similarity

→ “according to values of similarity”?

330 “spoken in the areas of Lugo, Mondonhedo and Ourense”

Consider adding a note on toponyms (since this is the orthographic form really used by Beswick 2007).

337 Fig. 6 has a small rectangle superimposed in the low left corner. It does not show the three standards...

334-338 In general the reasons why Fig. 5 differs from fig. 4 could be better commented.

Author Response

I gratefully appreciate the comments made by reviewers, which have greatly enriched the paper.

Point 1: 42-43 

Response: new version "the limitations of methods used in traditional dialectology"

Point 2: 49-50

Response: Using traditional methods (isoglosses) is really impossible to identify similarities, distances and correlations on a large set of data (conf. the cited works,  in particular Szmrecsanyi 2014, pp. 81-85).

Point 3: 130

Response: new version "of NORM speakers, as proposed by Chambers and Trudgill (1998)."

Point 3: 147

Response: rectified

Point 4: 230

Response: "by low-mobility adults, NORMs under the name proposed by Chambers and Trudgill (1998)."

Point 5: 238-244

Response: Data are presented following a logical order: i) source of data; ii) selection criteria.

Point 6: 273

Response: new version "Dialectometric techniques consider linguistic variation quantitatively, allowing to obtain information about the spatial stratification of the dialectal similarities as regards the standard"

Point 7: 274-275

Response: See 279-287, about Jenks classification method.

Point 8: 303 

Response: rectified

Point 9: 337

Response: Figure 6 has been rectfied and changed.

Point 10: 334-338

Response: Figures 4 and 5 result from application of different statistical techniques, as futher discussed in 3.2 and 4 sections.

Reviewer 2 Report

I would suggest to make plainly clear that the measure of distance is based only or mainly on morphosyntactic feaures.

Some corrections and suggestions are included in the the revised text. 

Author Response

I gratefully appreciate the comments made by reviewer, which have greatly enriched the paper.

I agree with all of the comments and suggestions, which have been incorporated into the new version.
